# Genome-Wide Identification, Evolution and Expressional Analysis of *OSCA* Gene Family in Barley (*Hordeum vulgare* L.)

**DOI:** 10.3390/ijms232113027

**Published:** 2022-10-27

**Authors:** Kuijun She, Wenqiu Pan, Ying Yan, Tingrui Shi, Yingqi Chu, Yue Cheng, Bo Ma, Weining Song

**Affiliations:** 1State Key Laboratory of Crop Stress Biology in Arid Areas, College of Agronomy, Northwest A&F University, Yangling, Xianyang 712100, China; 2Crop Research Institute, Ningxia Academy of Agriculture and Forestry Sciences, Yinchuan 750002, China

**Keywords:** barley, *OSCA* family, abiotic stress, genetic variation, expression pattern

## Abstract

The hyperosmolality-gated calcium-permeable channel gene family (*OSCA*) is one kind of conserved osmosensors, playing a crucial role in maintaining ion and water homeostasis and protecting cellular stability from the damage of hypertonic stress. Although it has been systematically characterized in diverse plants, it is necessary to explore the role of the *OSCA* family in barley, especially its importance in regulating abiotic stress response. In this study, a total of 13 *OSCA* genes (*HvOSCAs*) were identified in barley through an in silico genome search method, which were clustered into 4 clades based on phylogenetic relationships with members in the same clade showing similar protein structures and conserved motif compositions. These *HvOSCAs* had many cis-regulatory elements related to various abiotic stress, such as MBS and ARE, indicating their potential roles in abiotic stress regulation. Furthermore, their expression patterns were systematically detected under diverse stresses using RNA-seq data and qRT-PCR methods. All of these 13 *HvOSCAs* were significantly induced by drought, cold, salt and ABA treatment, demonstrating their functions in osmotic regulation. Finally, the genetic variations of the *Hv**OSCAs* were investigated using the re-sequencing data, and their nucleotide diversity in wild barley and landrace populations were 0.4966 × 10^−3^ and 0.391 × 10^−3^, respectively, indicating that a genetic bottleneck has occurred in the *OSCA* family during the barley evolution process. This study evaluated the genomic organization, evolutionary relationship and genetic expression of the *OSCA* family in barley, which not only provides potential candidates for further functional genomic study, but also contributes to genetically improving stress tolerance in barley and other crops.

## 1. Introduction

As sessile organisms, plants usually have to suffer from the impact of adverse abiotic stresses, such as drought, salinity and extreme high or low temperature [1,2], which not only determine the geographical adaptability of plants, but also seriously limits agricultural productivity worldwide [3]. To cope with these stresses, plants have evolved complicated and interconnected mechanisms to protect them from the impacts of harsh environments in a gradual manner [4]. Generally, plants can perceive and transduce extracellular stress signals into intracellular second messengers, which are able to trigger a serial of signal cascades to regulate the given metabolic and physiological processes through activating or inhibiting the expression of some specific genes, resulting in cellular homeostasis and the reprograming of growth to increase stress resistance or adapt to stressed conditions [5,6].

Calcium ions (Ca^2+^) mainly serve as crucial secondary messengers, playing a pivotal role in signaling perception and transduction under biotic and abiotic stresses, especially responding to osmotic change. Previous studies have revealed that calcium channels could act as osmosensors across unicellular prokaryotic bacteria to multicellular eukaryotic mammals [7,8,9,10,11]. In plants, several calcium channels have been characterized to find that they not only function to mediate the response to abiotic and biotic stresses, but are also involved in growth regulation and biosynthesis [12,13,14,15,16,17]. Among them, the hyperosmolality-gated calcium-permeable channel family (*OSCA*) is one of the most important and well-studied calcium channels participating in osmotic adjustment [18,19]. In light of its significance, the *OSCA* family has been identified in many plant species, including 15 *OSCA* members found in *A. thaliana*, 11 in rice [18], 42 in wheat [19], 12 in maize [20,21], 13 in mung bean [22], as well as 35 in cotton [23]. The expression profile of *OSCA* genes were also systematically investigated under diverse abiotic stresses in different plant species, which demonstrated their differential roles in stress response and tolerance [18,19,20,21,22,23]. Some plant *OSCA* genes were also functionally validated. In *A. thaliana*, *AtOSCA1.3* was phosphorylated by the immune receptor-associated cytosolic kinase BIK1 by treatment with pathogen-associated molecular patterns (PAMP) *fg22*, and showed an increase in channel activity, which functioned as a switch to control stomatal closure in the immune signaling pathway [24]. *At**OSCA1* was found to act as an osmosensor through a calcium imaging-based genetic screen approach, which was a plasma-membrane located protein with a low hyperosmolality-induced Ca^2+^ increase (OICI) [25]. In maize, *Zm**OSCA2.3* and *Zm**OSCA2.4* were up-regulated under ABA, PEG and NaCl treatments and over-expression of *Zm**OSCA2.4* could increase the expression of drought tolerance related genes while decreasing the expression of senescence-related genes in *A. thaliana*, respectively [21]. The expression profile analysis of *Zm**OSCAs* in 15 tissues found that *Zm**OSCA4.1* was shown to be up-regulated in all tissues and the genetic variations in *Zm**OSCA4.1* were significantly associated with drought tolerance at the seedling stage through association analysis [20]. Over-expression of *Os**OSCA1.4* in the *A. thaliana osca1* mutant can be complementary with its function in hyperosmolality reactions under salt stress [26]. Over-expression of eight *OsOSCAs* in *A. thaliana* demonstrated that these *OsOSCA* members were mainly involved in osmotic perception and stress adaption, which could regulate stomatal closure, leaf water loss and root growth through mediating osmotic Ca^2+^ signaling [27]. Additionally, the function of *Gh**OSCA1.1* was validated by a virus-induced gene silenced (VIGS) approach, which controlled the activity of antioxidant enzymes to enhance salt and drought resistance [23].

Barley (*Hordeum vulgare* L.) is the fourth largest cereal crop all over the world in terms of cultivation area and annual total yield, with great edible, forage and brewing value [28]. Furthermore, barley is also one of the most stress tolerant crops globally, such as salt, low temperature and soil infertility stress, providing the gene pool for genetic improvement against biotic/abiotic stress. However, it is necessary to explore the *OSCA* gene family further in order to mine the elite candidates regulating abiotic stress tolerance. The objective of the current study was to systematically identify the barley *OSCA* gene family (*HvOSCAs*) based on a genome-search approach and its genomic organization, exon-intron structures and phylogenetic relationship were investigated. Furthermore, the expression patterns of them were systematically detected during development stages and under diverse abiotic stresses, and a co-expression network of them was constructed based on RNA-seq data to obtain the stress-related modules. Finally, quantitative real-time PCR (qRT-PCR) analysis was used to validate their expressions under diverse stresses and the stress-related candidates were found. Additionally, the genetic variations of these *HvOSCAs* were investigated using population re-sequencing data.

## 2. Results

### 2.1. Identification of OSCA Gene in Barley

Based on a genome-search method [18,19], a total of 13 putative *OSCA* genes were identified in the barley genome, which were nominated as *HvOSCA1.1* to *HvOSCA4.1* according to the orthologs in *A. thaliana* and rice. As two orthologs of *OSCA1.3* and *OSCA2.1* were found in barley, they were named *HvOSCA1.3_1* and *H**vOSCA1.3_2*, *HvOSCA**2.1_1* and *HvOSCA**2.1_2*, respectively, indicating that segmental duplication events of *OSCA* genes had occurred in the barley genome. Sequence characteristics of these *HvOSCAs* showed that their gene length ranged from 2385 (*HvOSCA4.1*) to 26,794 (*HvOSCA1.1*) base-pair (bp), while their coding sequence length ranged from 2133 bp to 2400 bp, encoding 711 to 800 amino acids. The isoelectric point (pI) varied from 6.7 to 9.24, and all of the *HvOSCA* proteins had a positive GRAVY score, indicating that *HvOSCA* proteins were hydrophobic proteins. In addition, the subcellular localization prediction found that all of the *HvOSCAs* were located in the inner-membrane (Appendix A), which was consistent with previous studies [18,20,23]. Chromosome distribution analysis found that these 13 *HvOSCAs* were unevenly distributed on 5 out of 7 barley chromosomes (Appendix A), of which chr1H, chr4H and chr5H contained 3 *OSCA* members, chr2H and chr3H harbored 2 *OSCA* members, and chr6H and chr7H did not have any *OSCA* members (Appendix A).

### 2.2. Phylogenetic Relationship, Conserved Motifs and Gene Structure Analysis

To explore the phylogenetic relationships of the *OSCA* family, a neighbor-joining (NJ) phylogenetic tree was constructed using the full-length protein sequences of 13 *HvOSCAs*, 42 *TaOSCAs*, 15 *AtOSCAs*, 11 *OsOSCAs* and 11 *ZmOSCAs* (Appendix A). Results showed that these *OSCA* proteins were classified into four clades based on phylogenetic relationship (Figure 1), which was consistent with previous studies [18,19]. Further, thirteen *HvOSCAs* were distributed asymmetrically in each clade, with five, six, one and one members belonging to clade 1 to clade 4, respectively. In general, the *HvOSCAs* displayed closer phylogenetic relationships with those of wheat and rice compared to that of *A. thaliana*.

Then, the conserved motifs and exon-introns of the *HvOSCAs* were analyzed (Figure 2). It was indicated that all of the 13 *HvOSCAs* possessed three conserved function domains, RSN1_TM (PF13967), 7TM (PF02714) and PHM7_cyt (PF14703) (Figure 2B), which were the specific components of *OSCA* proteins [29,30], indicating the accuracy of the prediction of *HvOSCAs*. Furthermore, a total of 10 conserved motifs were also found in *HvOSCAs* (Figure 2C,E, Appendix A). The majority of *HvOSCAs* displayed the relatively consistent motif compositions with 8 to 10 conserved motifs, of which *HvOSCA2.1_1*, *HvOSCA2.1_2*, *HvOSCA2.2* and *HvOSCA2.3* contained 9 motifs except motif-6, while *HvOSCA1.1* and *HvOSCA3.1* contained 9 motifs except motif-1. Additionally, *HvOSCA4.1* had only one conserved motif. These different organizations of conserved motifs might correlate to the different function of these *HvOSCAs*.

Exon-intron structure is an evolutionary force to decide the functional diversification of the members in gene family [31]. It was found that the intron characteristics of *HvOSCAs* were highly variable with the number ranging from 0 to 11. Similar to its ortholog *OsO**SCA4.1* and *At**OSCA4.1*, *HvOSCA4.1* had no introns reported, while *HvOSCA1.1* had 11 introns. Additionally, *HvOSCA1.1* also had the longest intron length, which might due to a retro-transposon-like element insertion. Overall, the members in the same clade based on phylogenetic relationship shared similar motif compositions and exon-intron structure, suggesting that they might have more similar biological functions.

### 2.3. Cis-Element Analysis of HvOSCAs

A total of 37 kinds of cis-elements were predicted in the upstream 2000 bp from the transcription start sites of *HvOSCAs* (Figure 3, Appendix A), which were widely involved in growth biological process, hormone responsiveness, light responsiveness, metabolic regulation, as well as stress response. Among them, the elements associated with response to biotic or abiotic stresses were identified in almost all *HvOSCAs*, such as the drought responsive element (MBS) found in eleven *HvOSCAs*, anaerobic induction elements (ARE) in ten *HvOSCAs*, anoxic specific induction elements (GC-motif) in five *HvOSCAs*, as well as a wound responsive element (WUN-motif) in one *HvOSCA* and a defense-responsive element (AT-rich element) in one *HvOSCA*. Furthermore, nine types of hormone-responsive regulatory elements were also found, including auxin-responsive elements (AuxRR-core and TGA-element) in four *HvOSCAs*, gibberellin-responsive elements (P-box and GARE-motif) in eight *HvOSCAs*, salicylic acid-responsive elements (TCA-element) in four *HvOSCAs*, MeJA-responsive elements (CGTCA-motif and TGACG-motif) in ten *HvOSCAs* and ABA-responsive elements (ABRE) in eleven *HvOSCAs*. Additionally, there were meristem expression-related elements (CAT-box) in five *HvOSCAs*, circadian control-related elements (circadian) in two *HvOSCAs* and a cell cycle regulation-related element (SA-like) in one *HvOSCA*. The results provided some clues about the putative roles that *HvOSCAs* play in barley growth, development and stress response.

### 2.4. Expression Profiles of HvOSCAs in Tissues and under Stress Conditions Based on RNA-seq Data

To obtain some clues about the potential function of *HvOSCA**s*, spatio-temporal expression profiles of them were investigated based on previous open access available RNA-seq data of 16 different tissues at different growth stages (Figure 4A, Appendix A). Current study results showed that *HvOSCA**1.1* and *HvOSCA**3.1* were highly expressed in almost all tested tissues across diverse developmental stages, indicating that these *OSCAs* played important roles in regulating the growth and development of barley. *HvOSCA4.1* was found to be expressed in all tissues/stages but with a relative lower level. At the same time, six *HvOSCAs* displayed no expression in these tissues/stages, including *HvOSCA1.2, HvOSCA1.3_1*, *HvOSCA1.3_2*, *HvOSCA3.1*, *HvOSCA2.3* and *HvOSCA2.5*, suggesting that they might not function on barley growth and development. *HvOSCA1.4* showed high expression in CAR5, RAC and LOO, but no expression in other stages/tissues. *HvOSCA2.2* showed relatively higher expression in CAR15, LEM, NOD, RAC and ROC while lower expression in other tissues/stages. Furthermore, some tissue-specific *HvOSCAs* were also identified. It was interesting that the segmental duplication genes *HvOSCA2.1_1* and *HvOSCA2.1_2* showed diverging expression patterns in these tissues: *HvOSCA2.1_2* was highly expressed in EPI, ETI, LEA and NOD, while *HvOSCA2.1_1* only expressed in LEM and LOD, despite having highly conserved sequence organization. This divergent expression pattern might be due to their different cis-element compositions, and further study on their expression differentiation could provide helpful information reharding the evolution and function of the duplicated genes.

As hyperosmotic stress sensors, *OSCAs* play a vital role in regulating abiotic stress signaling. Thus, the expression profiles of *HvOSCAs* were further investigated under cold, salinity and heavy metal ion abiotic stresses using the previous open access RNA-seq data (Figure 4B, Appendix A). Under cold stress treatment, *HvOSCA1.2*, *HvOSCA1.3_1*, *HvOSCA2.3* and *HvOSCA2.5* did not have any expression, while *HvOSCA4.1* highlighted no differential expression between control and cold treatments. Compared to CK, only *HvOSCA1.1*, *HvOSCA2.1_1*, *HvOSCA2.1_2*, *HvOSCA1.3_2* and *HvOSCA2.4* showed up-regulated expression under cold stress treatment (Figure 4A), which could be considered as the cold-responsive candidates. Then, the expression patterns of *HvOSCA* genes were investigated in the meristem, elongation and mature zones of roots under salinity stress treatment, and most of them showed differential expressions (Figure 4B, Appendix A). Compared with CK (mock), a total of eight, eight and four *HvOSCA* genes were significantly up-regulated in the meristem, elongation and maturation zones of roots, respectively. Especially, *HvOSCA2.2* and *HvOSCA2.4* exhibited an increasing expression in all three zones relative to the control. Under metal ion stresses, *HvOSCA2.1_1*, *HvOSCA2.2* and *HvOSCA2.4* were up-regulated under copper (Cu)- and cadmium (Cd)-stressed conditions, while down-regulated under zinc (Zn) treatment. By contrast, *HvOSCA1.1*, *HvOSCA1.2*, *HvOSCA1.3_2* and *HvOSCA4.1* were down-regulated under Cu and Cd treatments but up-regulated under zinc treatment. *HvOSCA1.1* and *HvOSCA1.2* were down-regulated in all of the three heavy metal ion treatments. The metal ion-responsive *HvOSCAs* provided potential targets for further functional genomics study to reveal their roles in regulating osmotic adjustment and ion toxicity.

### 2.5. Co-Expression Network Analysis of HvOSCAs Involved in Abiotic Stress Response

To get more information about the function and molecular modules of *HvOSCAs* that are involved in the response to abiotic stresses, the WGCNA method was employed to analyze the co-expression network based on the RNA-seq data under diverse stresses, and the *HvOSCAs* associated modules were identified. The present study results showed that a total of 27,987 genes displayed significant expressions, and these were used for constructing co-expression networks. A total of 47 co-expression modules with the gene number ranging from 34 to 7981 were obtained (Appendix A). Then, the modules that 13 *HvOSCAs* were involved were further identified. Results found that 12 out 13 *HvOSCAs* were inv olved in 7 co-expression modules, including cyan, brown, blue, turquoise, green, pink and red modules. It is interesting that these 12 *HvOSCAs* had the conserved RSN1_7TM functional domain that is the most prominent characteristic of calcium-permeable stress-gated channel proteins, while the remaining *HvOSCA2.3* without the RSN1_7TM domain was not found in these constructed modules. Furthermore, a correlation analysis of these seven modules with the stress treatments was performed (Appendix A). Interestingly, one, one, four, one, two, one and one *HvOSCAs* were found in the cyan, brown, blue, turquoise, green, pink and red, respectively. The red module was significantly correlated with cadmium treatment and the turquoise module was correlated with cadmium treatment and copper treatment, respectively. The green module was correlated with NaCl treatment of the mature zone of the root, and also correlated with copper treatment. The brown module was significantly correlated with NaCl treatment of the elongation zone. The cyan and pink modules were both correlated with NaCl treatment of the mature zone. The results suggest the function of *HvOSCAs* in the heavy metal ion and salt response.

Then, the current study predicted the miRNA that interacted with *HvOSCAs* and was involved in the stress response. Through prediction, eight *HvOSCA* genes were found to be targeted by thirteen miRNAs, including miR156, miR6195 and so on. Among them, *HvOSCA1.3_1*, *HvOSCA1.1* and *HvOSCA1.1* could be bound by three miRNAs, respectively. Meanwhile, *Hvu-miR6206* could target both *HvOSCA1.3_1* and *HvOSCA1.2*, and *Hvu-miR6195* could target both *HvOSCA1.2* and *HvOSCA2.4.* Furthermore, with integration of the miRNA–*HvOSCA* relationship and the co-regulation modules of *HvOSCAs*, the sophisticated regulatory network that *HvOSCAs* use as the hub and is mediated by miRNAs was constructed, and a total of 16 miRNA-*HvOSCA* interactions were obtained (Figure 5, Appendix A). The miRNA-mediated networks contributed to better understand the roles of *HvOSCAs* in regulating stress response and resistance in barley, which paved a way to modulate *HvOSCAs* expression to induce some physiological changes and then enhance stress resistance through a post-transcriptional approach.

### 2.6. Validation of the Expression of HvOSCAs by qRT-PCR Assays

As *OSCA* genes function as hyperosmotic stress sensors, they play a vital part in abiotic stress response, hormone regulation and signaling transduction processes [25]. To identify the salt-responsive candidates, the expression profiles of all 13 *HvOSCAs* were systematically validated under drought, salt, cold and ABA treatments by qRT-PCR analysis (Appendix A). Under 100 µM ABA treatment, *HvOSCA2.1_1*, *HvOSCA2.1_2*, *HvOSCA2.4* and *HvOSCA3.1* showed up-regulated expression; in particular, the expression level of *HvOSCA2.1_1* was significantly high. *HvOSCA4.1* had gradually increased expression with the extension of treatment, and reached the peak at 24 h (h) after treatment. Moreover, the expressions of *HvOSCA2.1_2*, *HvOSCA2.4* and *HvOSCA3.1* were significantly up-regulated in the first 6 h, but decreased significantly as the treatment time continued, and finally were significantly lower compared to CK. Contrary to these results, the expressions of *HvOSCA1.1*, *HvOSCA1.4* and *HvOSCA2.2* displayed a continued decreasing trend, and reached the lowest levels at 24 h.

Under cold treatment, *HvOSCA1.1*, *HvOSCA2.1_1*, *HvOSCA2.1_2*, *HvOSCA2.2*, *HvOSCA3.1* and *HvOSCA4.1* were found to exhibit a remarkably higher expression level in the first 6 h and then decreased at 12 h after treatment. The expression level of *HvOSCA1.3_2* was gradually increased with the duration of treatment, and reached the peak at 12 h, and then decreased at 24 h. The expression of *HvOSCA1.4* showed no significant difference in the first 6 h of treatment, and then decreased significantly at 12 h, and finally increased sharply at 24 h. These results were consistent with the trends found by RNA-seq data and the differential expressed genes provided the candidates associated with the cold response.

Under 20% PEG-6000 treatment, these *HvOSCAs* showed completely diverse expression patterns. The expressions of *HvOSCA1.1* and *HvOSCA1.4* showed consistence with that of CK at 6 h of drought treatment, and increased rapidly to reach the highest level at 12 h. Then, *HvOSCA1.1* maintained a high expression at 24 h of treatment, while *HvOSCA1.4* decreased remarkably at 24 h. *HvOSCA1.3_2* displayed the highest expression level at 6 h, and then decreased gradually during the drought treatment. At the same time, *HvOSCA2.1_1*, *HvOSCA2.2* and *HvOSCA4.1* showed similar expression patterns that were significantly down-regulated at 6 h and 12 h of treatment, and then up-regulated at 24 h. *HvOSCA2.1_2* seemed to function as a negative regulator of the drought response and displayed significantly down-regulated expression at 12 h. Finally, the expression levels of *HvOSCA2.4* and *HvOSCA3.1* increased at 6 h, then decreased sharply at 12 h, but increased significantly to the highest level at 24 h. The different expression patterns of these *HvOSCAs* suggest their divergent regulatory roles in drought response and tolerance.

Under salt stress treatment, these *HvOSCAs* also showed obviously different expression patterns. *HvOSCA2.1_2* displayed the up-regulated expression all the time with a high expression level, while *HvOSCA2.2* and *HvOSCA4.1* showed down-regulated expressions at all time courses, suggesting that these might function in differential stages of salt stress. *HvOSCA3.1* and *HvOSCA2.4* were up-regulated at both 6 h and 12 h, while down-regulated at 24 h. The expression levels of *HvOSCA1.1* and *HvOSCA1.4* was the highest at 6 h, then decreased sharply at 12 h, but increased a little at 24 h. *HvOSCA1.3_2* showed no differential expression with CK at 6 h, but was up-regulated at 12 h and then down-regulated at 24 h. The expression of *HvOSCA2.1_1* decreased rapidly after salt treatment, and reached the lowest expression at 6 h, then increased steadily to the highest level at 24 h. The results provided some clues on their divergent function on salt response and tolerance. Interestingly, the expressions of *HvOSCA1.2*, *HvOSCA1.3_1*, *HvOSCA2.3* and *HvOSCA3.1* had extremely low or seldom expression under all stress conditions, which was also demonstrated by RNA-seq analysis, indicating they might not function in the stress response (Figure 6, Appendix A).

### 2.7. Genetic Variations and Haplotype Analysis of HvOSCA

The genetic variations of these *HvOSCA* genes were investigated based on the resequencing data of wild barley and cultivated barley populations, including 85 wild and 135 landrace accessions. A total of 2077 SNP loci were found in the 13 *HvOSCA*s and then were used to calculate the population divergence (*F*st) and nucleotide diversity (π) values. The current study results found that the *Fst* value of them ranged from 0 to 0.808825. According to a previous study, the *F*st value of a gene exceeds the threshold line (*F*st = 0.456) was considered as an artificially selected gene. Among these 13 *HvOSCAs*, *HvOSCA1.3_1*, *HvOSCA2.2* and *HvOSCA1.1*, as well as *HvOSCA2.2*, were under artificial selection during barley domestication (Figure 7A, Appendix A).

Furthermore, the genetic diversity of *HvOSCAs* in the wild barley population was 0.4966 × 10^−3^, while that of landrace population was 0.391 × 10^−3^ (Figure 7B, Appendix A). Compared to wild barley, the cultivated barley had relatively lower genetic diversity, suggesting that genetic bottleneck had occurred on the barley *OSCA* family during the barley evolution process. Further, the current study observed that the haplotype frequencies of the *HvOSCAs* were investigated based on the SNP variations. Results found only four genes showed different haplotype frequencies between wild and cultivated barley, and all of them displayed more haplotypes in wild barley than that of cultivated barley, which also demonstrated the genetic bottleneck effect on the *OSCA* family when wild barley domesticated to a landrace (Figure 7C, Appendix A).

### 2.8. Subcellular Localization of HvOSCA1.1, HvOSCA 2.4, HvOSCA3.1 and HvOSCA 4.1

In order to validate the subcellular localization of *HvOSCA* proteins, four *HvOSCAs* genes, including *HvOSCA1.1, HvOSCA2.4, HvOSCA3.1* and *HvOSCA4.1* were cloned from the barley cv. Morex to construct their corresponding transient vectors, and then separately injected into tobacco leaf cells to track the localization of the *HvOSCA* proteins through laser confocal microscopy. As shown in Figure 8, all four proteins were expressed as *HvOSCA*-GFP fusion protein in transformed tobacco leaf cells, and they were mainly localized in the inner-membrane.

## 3. Discussion

Plants inevitably undergo changing environments including adverse abiotic environments, such as a drought, soil salinity and extreme temperatures, throughout their life cycle, and they have evolved interconnected regulatory pathways in response to the exogenous and endogenous osmotic changes [3]. Barley ranked as the fourth most important cereal crop worldwide [32], which is widely distributed from the vicinity of Dead Sea in the Middle East where *H. vulgare* originated, to the high altitude areas of the Qinghai-Tibet plateau [33]. Barley has a wide range of adaptability and can withstand harsh environments [34]. Extensive studies have demonstrated that barely is one of the most stress tolerant staple crops all over the world with the excellent salt tolerance, drought and low temperature resistance, as well as adaptation to soil infertility, providing a gene pool for genetic improvement and breeding of stress tolerant crops. Therefore, mining elite genes underlying stress tolerance in barley will provide a useful gene resource for stress-oriented genetic improvement and breeding. The *OSCA* family acts as an osmosensor, playing a crucial role in the sensing of osmotic stresses and regulating osmotic adjustment in plants [25,30]. In view of its importance, the *OSCA* family has been systematically identified in several species, including *A. thaliana*, rice, maize and wheat, but not so far in barely. In this study, 13 *HvOSCA* genes were identified at genome level in barley. Based on phylogenetic analysis, these *HvOSCAs* could be clustered into four clades, and the members clustered in the same clade possessed the detected similar conserved motif composition, exon-intron structures and cis-element organizations, suggesting that they had similar function, which is consistent with previous studies [20]. Domains of the late exocytosis TM putative phosphate transporter and calcium-dependent channel of *OSCA* genes were found to be extremely conserved in higher plants [25]. All *HvOSCA* genes identified herein contained these three functional domains, further supporting the high conservation of the *OSCA* family and also demonstrating the accuracy of our prediction. 

The expression pattern is the external embodiment of gene function, and gene expression is largely regulated by the cis-elements in the promoter region [35]. Cis-regulatory elements play an essential role in a gene’s spatio-temporal expression, and then in regulating plant growth, development, morphogenesis, senescence, apoptosis, as well as in coordination and adaptation to the environment [36]. The current study found that *HvOSCAs* harbored various cis-elements related to growth, hormone responsiveness and also stress responsiveness. Combined with the expression profiles of *HvOSCAs* from RNA-seq and qRT-PCR analysis, the current study found that the cis-elements could regulate the expressional specificity of some *HvOSCAs*. For example, *HvOSCA1.3_2*, harboring five MBS cis-elements that are regulatory elements serving as a MYB binding site involved in drought inducibility, showed a quick response under PEG-induced drought treatment. *HvOSCA1.2_2* showed rapid up-regulation expression under ABA treatment and maintained at a high expression level, which contained seven CGTCA-motif elements that involved in the MeJA responsiveness. The current study further reconstructed the co-expression network by the WGCNA method based on public RNA-seq data, and further investigated the correlations between seven modules and simulated abiotic stress treatment including cold, NaCl and heavy metal ion (cadmium, copper and zipper) treatment. Results showed that seven modules were significantly associated with stress treatment. All modules harbored the *HvOSCA* genes as the hub factors. Furthermore, prediction results of miRNA regulating *OSCA* genes in seven modules showed that thirteen microRNAs were found to target eight *HvOSCA* genes involved in the stress-related modules, including *Hv*u-miR5049f, *Hv*u-miR6195, *Hv*u-miR6196 and *Hv*u-miR156a. Previous studies have been reported that *Hv*u-miR5049f could respond to drought stress in Tibetan wild barley [37], *Hv*u-miR6195 and *Hv*u-miR6196 were detected in rice root under salt treatment [38], and the expression of *Hv*u-miR156a was modulated by barley leaf stripe (BLS) in Tibetan barley [39]. We postulated that these four miRNAs might play a regulatory role in the response to stress by controlling the expression of *OSCA* genes in barley.

In previous studies, different *OSCA* members mainly functioned as stress responsive proteins in dehydration stress or acted as hyperosmolality sensors, and also played the role of a switch to control the influx of Ca^2+^ in the pathway of immune signal transduction [20,25,26,27,40]. The present study found that *HvOSCA2.1_1* was significantly induced by ABA and cold, but was inhibited by drought and salt treatment during the initial 12 h, while there was little change in expression of *HvOSCA4.1* under ABA, PEG, salt and cold (except 6 h) treatment compared with control. *HvOSCA2.2* was significantly inhibited by drought, salt and ABA, but markedly induced by cold in the initial 6 h. Among the cis-elements of *HvOSCA* genes, the number of hormone-responsive and stress-responsive elements were more than that of growth and metabolic-responsive elements, signifying that *HvOSCAs* mainly functioned in stress response and signal transduction.

Population genomic research has provided a crucial approach for the investigation of genetic divergence, domestication and evolution, as well as the genetic basis for the trait in plant populations [41,42,43,44].

The current study evaluated the genetic variations of the *Hv**OSCA* family in wild barley and landrace populations. *F*st values among different populations indicated that artificial selection had occurred in barley *OSCA* genes at different degrees between members of four identified *Hv**OSCAs* . The results showed that wild barley possessed higher nucleotide diversity and alternative haplotypes compared to landrace populations, which suggested that genetic bottleneck had occurred within this family during barley domestication. These results suggested that barley domestication occurred from the perspectives of the *OSCA* family, and are promising enrich the genetic diversity for barley breeding against abiotic stress resistance.

Generally, the subcellular localization helps to decipher its cellular function. Previous studies have reported that *OSCA* proteins were localized in the cell membrane in other plants [18,45]. In the current study, both the in silico prediction and the subcellular localization results indicated that the *HvOSCA* family members are localized in the inner-membrane. The results in the current study testified the previous conclusions and provided useful information for further functional studies of *HvOSCAs*.

## 4. Materials and Methods

### 4.1. Identification of OSCA Genes in Barley

For the identification of the *OSCA* gene family in barley, the protein sequences of the barley genome were retrieved from the Ensembl plant database (http://plants.ensembl.org/Hordeum_vulgare/Info/Index, accessed on 10 January 2022). The Hidden Markov Model (HMM) profile of homeodomain (HD) (DUF221) (renamed from DUF221 to RSN1_7TM (PF02714) was used as a query to search against the local protein database using the HMM3.0 tool with the E-value ≤ 1 × 10^−10^ as the threshold. Furthermore, 15 *At**OSCAs* and 11 *Os**OSCAs* protein sequences were used to perform a BLASTP search against the local protein database with the threshold of E-value < 1 × 10^−5^. Subsequently, the results of the HMMER and BLASTP searches were integrated together, and the redundants were manually removed to obtain the putative barley *OSCA* proteins. These data were submitted to the Conserved Domain Database (CDD) (https://www.ncbi.nlm.nih.gov/cdd, accessed on 10 January 2022), the Simple Modular Architecture Research Tool (SMART) (http://smart.embl-heidelberg.de/, accessed on 5 February 2022) and PFAM (https://pfam.xfam.org, accessed on 1 March 2022) to further confirm their *OSCA* conserved domain; those with a complete *OSCA* domain remained as candidates. Finally, the candidates were submitted to the ExPASy (http://web.expasy.org/protparam/, accessed on 1 March 2022) database to investigate the physicochemical properties such as molecular weight (Mw), isoelectric point (pI), instability index (II), aliphatic index (AI) and grand average of hydrophobicity (GRAVY). The subcellular localization of them was predicted by the online CELLO tool (v2.5) (http://cello.life.nctu.edu.tw/, accessed on 1 March 2022).

### 4.2. Phylogenetic Relationships, Genic Structure, Conserved Motif and Cis-Element Analysis

The protein sequences of the identified barley and previously reported *At**OSCA*s*, ZmOSCAs, OsOSCAs* and *TaOSCAs* were employed to construct the phylogenetic tree. Multiple sequence alignment was performed using Phylip software [46] and the MEGA-X tool was used for a phylogenetic tree constructed by the neighbor-joining (NJ) method with the bootstrap of 1000 replications. The chromosome location and exon-intron structures of these *HvOSCA* genes were retrieved according to the genome annotation files (http://plants. ensembl.org/Hordeum_vulgare/Info/Index, accessed on 10 January 2022) and then were visualized using MapGene2Chromosome v2.0 (http://mg2c.iask.in/mg2c_v2.0/, accessed on 1 March 2022) and Gene Structure Display Server (GSDS2.0) (http://gsds.cbi.pku.edu.cn/, accessed on 1 March 2022), respectively. The conserved protein motifs were predicted based on online MEME tools (http://alternate.meme-suite.org/, accessed on 1 March 2022) with the following parameters: the maximum number of motifs was set to 10, any number of repetitions was allowed, and the optimum width ranged from 6 to 250.

### 4.3. Cis-Element Analysis

The 2.0 kb genomic sequences upstream of the transcription start site of *HvOSCAs* were extracted from the barley reference sequence by in-home Perl script and submitted to the PlantCARE database (http://bioinformatics.psb.ugent.be/webtools/plantcare/html/, accessed on 12 March 2022) to predict the supposed cis-acting transcriptional regulatory elements.

### 4.4. Expression Profile Analysis of HvOSCAs Based on RNA-seq Data

A total of 142 RNA-seq data from the samples of different tissues at different development stages (including embryos, bracts, epidermis, young inflorescences, lemma, lodicule, shoot, rachis, developing tillers, palea, roots and leaves) and under diverse biotic and abiotic stresses (including drought, salinity, cold and ABA treatment) were downloaded from the NCBI Sequence Reading Archive (SRA) database (Appendix A). The HISAT2 (v2.1.0) [47] and StringTie (v1.3.5) pipeline [48] were used to calculate the value of fragments per kilobase per million (FPKM). The expression profiles of *HvOSCA**s* were retrieved and visualized with heat maps by using the pheatmap package in R software.

### 4.5. Co-Expression Network Analysis of HvOSCAs under Stressed Conditions

The R package WGCNA (V1.71, R 4.2) [49] was employed to conduct co-expression analysis based on the FPKM values matrix from various stresses with the default parameters except soft power = 8, min_module_size = 30, ME_miss_thread = 0.2. The FPKM values matrix of cold, NaCl and heavy metal ion (cadmium, copper and zipper) treatment were used for correlation analysis, and visualization was done by the ggcor package (V0.9.8.1). Thirteen *HvOSCA**s* protein sequences were used as queries to predict the miRNA binding sites by using published miRNAs of barley in the psRNATarget database (https://www.zhaolab.org/psRNATarget/analysis?function=2, accessed on 18 May 2022) with the default parameters. The software Cytoscape v3.9.1 [50] was employed to visualize the regulatory network with miRNA of *HvOSCA**s*.

### 4.6. Plant Materials Preparation, Stress Treatment and qRT-PCR Analysis

The seeds of barley cv. Morex, which is stored by the Wheat Genomics Lab in Northwest A&F University (Central Shaanxi Province, China; 34 26′ N latitude, 108 07′ E longitude, altitude: 458 m), were sterilized by 5% (*w/v*) sodium hypochlorite (NaClO) (Xinyicheng, Shaanxi, China) for 5 min, and then rinsed using distilled water for 3 min, and finally incubated on wet filter paper in a petri dish at 25 °C for 5 days [51]. This experiment was conducted from 10 to 30 March 2022 in NWSUAF. When the seedlings had grown to 4–5 cm, they were transplanted to hydroponic colonization plates, and put in culture medium (aqueous solution with 1/2 Hogeland nutrient) to reach the trifoliate leaf stage in a growth chamber under 20 °C /15 °C day/night, 16 h light/8 h dark cycle, and 50% relative humidity. Hydraulically cultured trifoliate leaf stage seedlings were exposed to culture medium, adding 150 mM NaCl, 20% PEG, 4 °C, or 100 µM ABA, and the roots and leaves were sampled at 0, 6 h, 12 h and 24 h as the stressed treatments [51,52]. The seedlings grown in the culture medium without any treatment were sampled simultaneously at the corresponding time point as the control. All the samples were collected from at least three replicates, frozen by liquid nitrogen immediately and then stored at −80 °C. Total RNA was extracted by Plant RNA Kit reagent (Omega BioTek, Norcross, GA, USA), then synthesis cDNA by 5× All-in-one RT MasterMix (ABM,Vancouver, Canada) following the manufacturer’s protocols. The *HvACTIN* (HORVU.MOREX.r2.5HG0378970) was used as the reference gene. The relative expression of *HvOSCA*s was quantified using the TB-Green ^®^Premix Ex Taq™ II kit (Takara, Dalian, China) and ABI7500 Real-Time PCR Systems (Applied Biosystems, Foster City, CA, USA). The qRT-PCR thermal cycling condition as follows: 95 °C for 30 s, followed by 40 cycles of 15 s at 95 °C, and 30 s at 60 °C. The 2^−ΔΔCT^ quantification method was used for calculating the relative expression level [52,53]. Three parallel technical replicates were set for each sample. The primers used in the current study are listed in Appendix A.

### 4.7. Genetic Variations and Haplotype Analysis of HvOSCAs

The resequencing data of 220 genotypes of barley (including 135 landraces and 85 wild barley) were obtained by an exome-captured approach from the NCBI SRA database (PRJEB8044/ERP009079), producing a visual component framework (VCF) file [43,52]. The SNPs of *OSCA* genes subsets were retrieved from the VCF file. The fixation index (*F*st) was analyzed by vcftools v0.1.16 with the parameters as window = 100, step = 1, and visualized by R package CMplot. The nucleotide diversity (π) was also calculated by using vcftools v0.1.16, and visualized by R package ggsci. The variant of all *HvOSCAs* were annotated by snpEff software, the intro-variants were filtered, and the protein-coding variants were used for haplotype analysis by in-home python scripts [54], and visualized by R packages ggsci and scales.

### 4.8. Subcellular Localization of HvOSCA-GFP Fusion Proteins

One *HvOSCA* member from each clade based on phylogenetic relationships (*HvOSCA1.1*, *HvOSCA 2.4*, *HvOSCA3.1* and *HvOSCA4.1*) was randomly selected for amplification using the primers listed in Appendix A. The PCR products were separately recovered and cloned into the multiple cloning sites of *Nde*I and *Sac*I in the pBI121-GFP vector. Each recombinant vector was separately transformed into *Agrobacterium tumefaciens* strain GV3101 prepared for agro-infiltration. For transient expression of HvOSCAs in tobacco, the Agrobacterium strains harboring the recombined vector were injected into 4-week-old tobacco leaves. Then, the images of the leaves were captured after 48 h of agro-infiltration by a laser confocal microscope (Olympus, Tokyo, Japan).

## 5. Conclusions

The current study identified and characterized a new *OSCA* gene family in barley with 13 *HvOSCAs* belonging to 4 clades. The exon-intron structures analysis found that they displayed highly variable intron numbers ranging from 0 to 11. Expression profiles and qRT-PCR validation found that these *HvOSCAs* were significantly induced by drought, cold, salt and phytohormone ABA treatment, demonstrating their function in stress response. Finally, genetic variations and haplotype analysis indicated that artificial selection was conserved on the *OSCA* family during wild barley domestication to the barley landrace. This study not only paves the way for further elucidating the function of *HvOSCAs* on stress response and tolerance but also provides a potential gene resource for breeding studies on stress tolerance in barely and other crops.

## Figures and Tables

**Figure 1 ijms-23-13027-f001:**
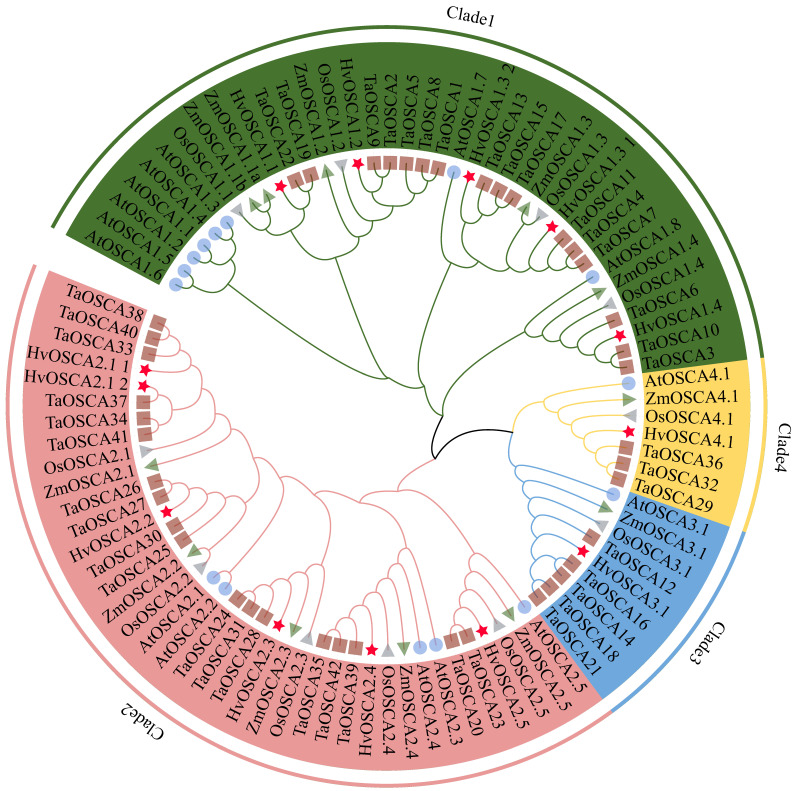
Phylogenetic analysis of the *OSCA* genes in *Hordeum vulgare* (*Hv*), *Arabidopsis thaliana* (At), *Triticum aestivum* (Ta), *Oryza satival* (Os) and *Zea mays* (Zm) by the neighbor-joining method. The *OSCAs* were clustered into four clades; each member of the *OSCAs* were annotated by (★ for *Hv*), (● for At), (■ for Ta), (◄ for Os) and (► for Zm), respectively.

**Figure 2 ijms-23-13027-f002:**
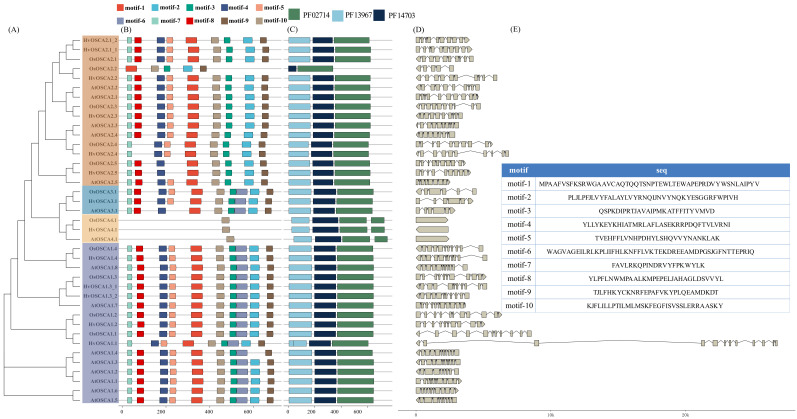
Motif composition, conserved domain and gene structure analysis of *HvOSCA*s. (**A**) Phylogenetic relationship of *HvOSCAs*, members of different clades were marked in different background colors. (**B**) Functional motif compositions of these 13 *HvOSCAs.* Ten motifs were represented by rectangle box in different colors. (**C**) The function conserved domain of PFAM02714, PFAM13967 and PFAM14703 organization in *HvOSCAs*, represented by dark blue, dark turquoise boxes and purplish blue, respectively. (**D**) Intron-exon organizations of *HvOSCAs*. The introns and exons are represented by the broken line and gray boxes, respectively. (**E**) The protein sequence of the 10 identified motifs.

**Figure 3 ijms-23-13027-f003:**
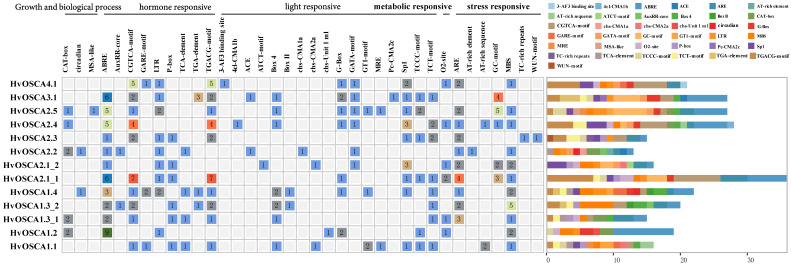
Cis-elements found in the promoter region of *HvOSCAs*. (**Left**): The number and function classification of cis-acting element in each *HvOSCA* genes. (**Right**): Distribution of 37 identified cis-acting elements in each *HvOSCA*; elements are represented by the boxes in different colors.

**Figure 4 ijms-23-13027-f004:**
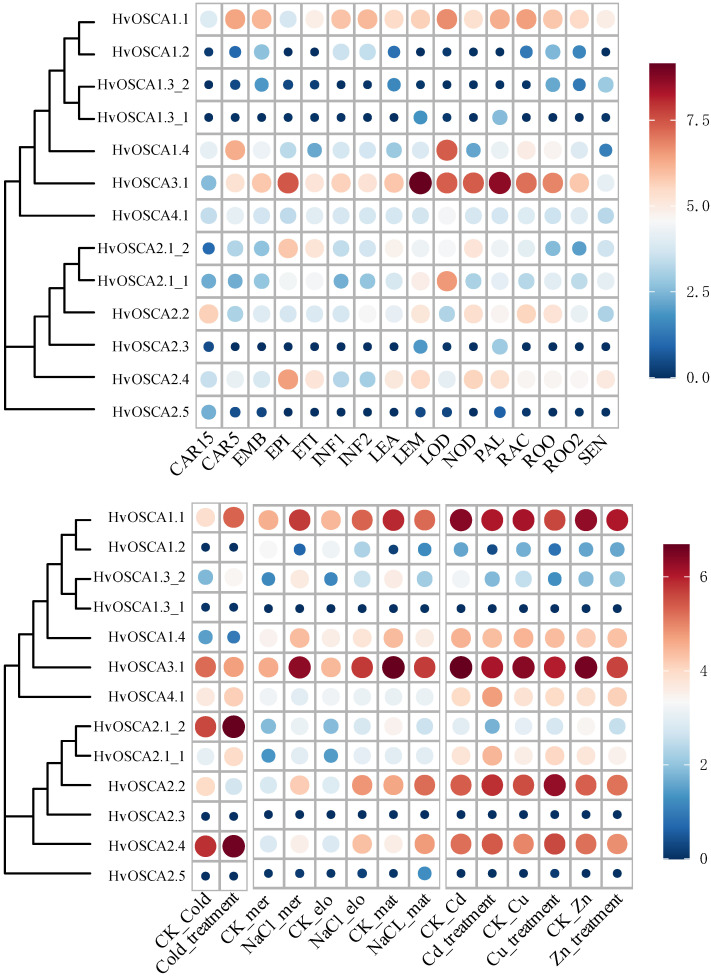
Expression profiles of *HvOSCAs* in different tissues and under diverse stresses. (**A**) The spatio-temporal expression profile of *HvOSCAs* in different tissues at different developmental stages. FPKM values were normalized by log_2_(FPKM+1) transformation to display the heatmap color scores. CAR15: bracts removed grains at 15DAP; CAR5: bracts removed grains at 5DAP (CAR5); embryos dissected from 4-day-old germinating grains (EMB); epidermis at 4 weeks old (EPI); etiolated from 10-day-old seedling (ETI); young inflorescences with 5 mm (INF1); young inflorescences with 1–1.5 cm (INF2); shoot with the size of 10 cm from the seedlings (LEA); lemma at 6 weeks after anthesis (LEM); lodicule at 6 weeks after anthesis (LOD); developing tillers at the six-leaf stage (PAL); 6-week-old palea (NOD); rachis at 5 weeks after anthesis (RAC); root from 4-week-old seedlings (ROO2); roots from the seedlings at the 10 cm shoot stage (ROO). (**B**) Expression profile of *HvOSCAs* under cold, NaCl and heavy metal ion (Cd, Cu and Zn) conditions. Cold stress treatment: CK of cold (CK_cold), cold (4 °C) treatment (Cold_treatment), salt treatment of barley root (dissected and sampled the barley according to the organizational structure of root for meristem zone, elongation zone and mature zone). CK of meristem zone (CK_mer), meristem zone under salt treatment (NaCl_mer), CK of elongation zone (CK_elo), elongation zone under salt treatment (NaCl_elo), CK of maturation zone (CK_mat) and maturation zone under salt treatment (NaCl_mat). Heavy metal ion (Cd, Cu and Zn) stress. Control of cadmium ion stress (CK_Cd), cadmium ion treatment (Cd_treatment), control of copper ion treatment (CK_Cu), copper ion treatment (Cu_treatment), control of zinc ion stress (CK_Zn) and zipper ion treatment (Zn_treament).

**Figure 5 ijms-23-13027-f005:**
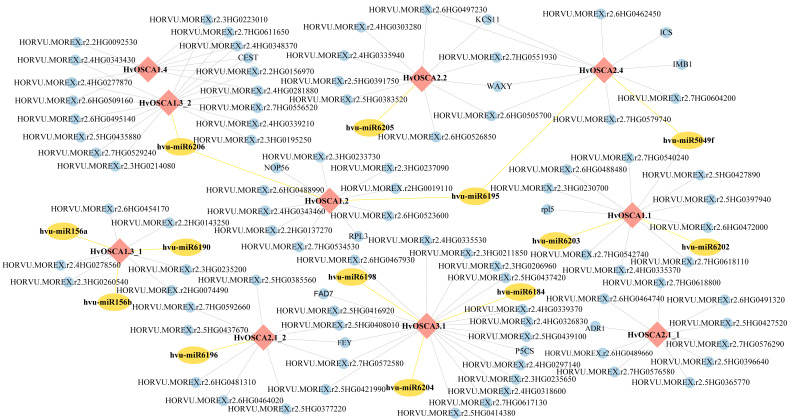
miRNA¬–mRNA interaction network of *HvOSCAs* involving responses to abiotic stresses. The *HvOSCAs* are represented by the red diamonds, the predicted miRNAs by yellow ovals and the other interacting genes by blue dots.

**Figure 6 ijms-23-13027-f006:**
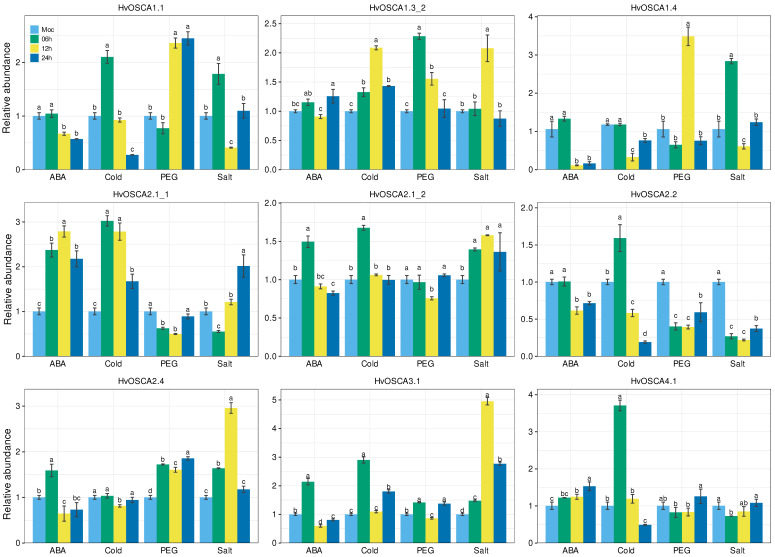
Validation of the expression signature of nine *HvOSCA* genes under ABA (100 µM ABA), cold (4 °C), drought (20% PEG-6000) and salt (NaCl) treatment through the qRT-PCR method. The treatments were conducted with 100 µM ABA solution for ABA, 4 °C for cold, 20% PEG 6000 solution for drought, and 150 mM NaCl solution for salt stresses, respectively. Data represent the average of three technical replications and the error bar represents the standard error of mean (SEM).

**Figure 7 ijms-23-13027-f007:**
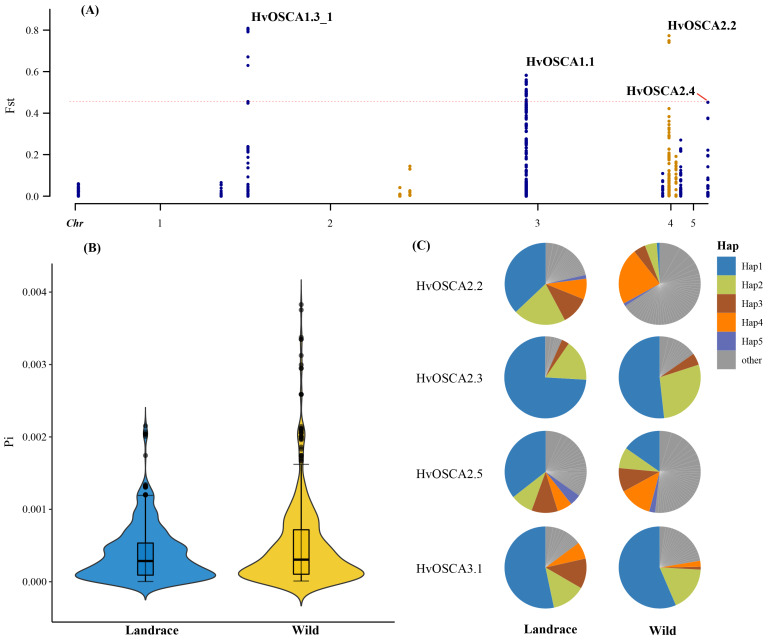
Genetic variations and haplotype analysis of *HvOSCAs* based on resequencing data. (**A**) Genetic divergence and artificial selection analysis of *HvOSCAs*; (**B**) nucleotide diversity of *HvOSCA* genes in landrace and wild barley populations; (**C**) haplotype frequency analysis of *HvOSCAs* in landrace and wild barley populations.

**Figure 8 ijms-23-13027-f008:**
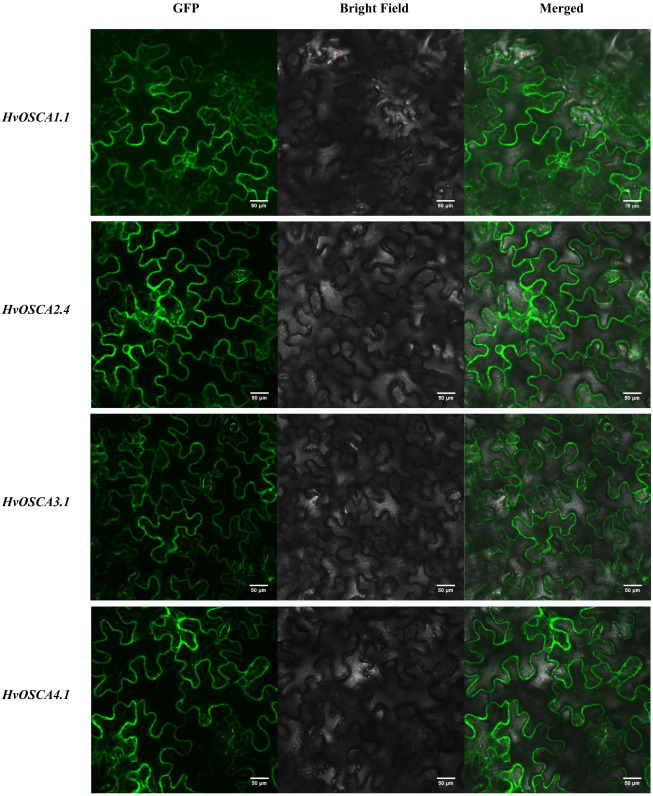
Subcellular localization of four recombinant CaMV35S: *HvOSCA*-GFP *HvOSCA* protein fusions transiently expressed in tobacco leaf cells.

## Data Availability

All of the datasets supporting the results of this article are included within the article and its additional files.

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
