# Peer review of "Genome-Wide Identification, Evolution and Expressional Analysis of OSCA Gene Family in Barley (Hordeum vulgare L.)"

_ijms, 2022, doi:10.3390/ijms232113027_

Round 1

Reviewer 1 Report

The manuscript entitled “Genome-wide identification, evolutionay and expressional analysis of OSCA gene family in barley” reports the identification of OSCA gene family in barley. The authors identified 13 members and they describe some analysis of the predicted sequences and some expression analysis in different tissues and under different treatments. Some comments and suggestions are listed in the following.

Main comments:

(1)  Identification and analysis of OSCA gene families has been performed in other plant species. The information provided by this manuscript does not provide enough information to the general knowledge about the characteristics of OSCA gene family.

(2)  In my opinion, the contents about the characteristics of OSCA gene family in the manuscript is very descriptive, it should be further improved.

(3)  I found, there is no obvious correlation between the results of available RNA-seq data and qRT-PCR analysis. The RNA-seq should be done using the same samples of barley with qRT-PCR experiments.

Minor comments:

(1)  Is the spell of word “evolutionay” in the title of manuscript correct?

(2)  The writing quality should be improved. For example, Page 9 Line 299, “Under 20% PEG-6000 treatment, these HvOSCA showed completely diverse expression patterns”,  “these HvOSCA” should be “these HvOSCAs” ?

(3)  Several figures (Figures 2, 3, and 5) are not clear. Draw them again.

Author Response

Question1:  Identification and analysis of OSCA gene families has been performed in other plant species. The information provided by this manuscript does not provide enough information to the general knowledge about the characteristics of OSCA gene family.

Response: We appreciated the reviewer’s professional and insightful comments on our manuscript. We agreed that the identification and analysis of OSCA gene families has been extensively performed in many plant species, such as Arabidopsis thaliana, rice, maize, cotton and wheat. Although OSCA genes displayed high conservation in plants, some differentiation was also found on the gene number, gene structure and expression patterns as well as biological function of OSCA family among different plant species. Thus, we conducted the genome-wide in silico search of OSCA family in barley in this study with the purpose to understand the characteristics of barley OSCA genes and also mine some targets for further functional study. In addition to the general well-established analysis of gene family, we also investigated the genetic variations, miRNA targeting and co-expression network of barley OCSA genes. To the best of my knowledge, these studies are not performed in other OSCA family analysis, which contributed to mine the excellent candidates and also better understand the characteristics of barley OSCA family. As commented, we have carefully revised the manuscript to highlight these results to provide some novel information to enrich the general knowledge about the characteristics of OSCA gene family.

Question2:  In my opinion, the contents about the characteristics of OSCA gene family in the manuscript is very descriptive, it should be further improved.

Response: We appreciated the reviewer’s professional and constructive suggestion on our manuscript. We have carefully improved the contents of Results section through giving comprehensively explaining the meaning of these results and also polish the contents of Discussion section by comprehensively referring to previous studies.

Question3:  I found, there is no obvious correlation between the results of available RNA-seq data and qRT-PCR analysis. The RNA-seq should be done using the same samples of barley with qRT-PCR experiments.

Response: We appreciated the reviewer’s professional and constructive suggestion on our manuscript. As the RNA-seq data was downloaded from public database, we can’t obtain the same samples used for RNA-seq analysis to perform qRT-PCR experiments. In this study, we adopted the same genotype barley cv. Morex and also under the same treatment used for RNA sequencing to conduct qRT-PCR experiments. We admitted that there were some differences between the results of RNA-seq data and qRT-PCR analysis. But the expression trend showed relative consistence between them. Additionally, to assure to obtain the reliable candidates, we validated the expression profiles of all 13 identified barley OSCA genes by qRT-PCR analysis and we selected the candidates mainly depending on the results of qRT-PCR analysis.

Minor comments:

(1)  Is the spell of word “evolutionay” in the title of manuscript correct?

 Response: We are sorry for this spelling error. We have revised it into “evolution”.

(2)  The writing quality should be improved. For example, Page 9 Line 299, “Under 20% PEG-6000 treatment, these HvOSCA showed completely diverse expression patterns”,  “these HvOSCA” should be “these HvOSCAs” ?

 Response: We appreciated the reviewer’s constructive and insightful comments. We have carefully revised this grammar error and also corrected the grammar errors and typos throughout the manuscript.

(3)  Several figures (Figures 2, 3, and 5) are not clear. Draw them again.

Response: We appreciated the reviewer’s constructive and insightful comments. We have redrawn the figures and provided figures in PDF format.

Reviewer 2 Report

This paper systematically reported the genomic organization, evolutionary relationship, genetic variation and expression profile of OSCA family in barley, which not only provided the potential candidates for further functional study, but also contributed to genetically improving for stress tolerance in barley and beyond.  Some minor problems are as follows:

1.In line35,between with and stress, an extra space.

2.What is the subcellular localization of this OSCA family genes in other species. A few genes can be picked for verification.

3.line 135, HvOSCA2.1_2 is repeat.

4.In FIG2A, different colors to different clades gene name can help reader distinguish phylogenetic relationship.

5.The number of OSCA genes described in line 267 is not consistent with that in Figure 5. And in Figure 5, what do the blue dots represent? The meaning and function of Figure 5 should be explained in more detail.

Author Response

1.In line35,between with and stress, an extra space.

Response: We have fixed it.

2.What is the subcellular localization of this OSCA family genes in other species. A few genes can be picked for verification.

Response: We appreciated the reviewer’s constructive and insightful comments. We have added the results of subcellular localization of four barley OSCA genes.

3.line 135, HvOSCA2.1_2 is repeat.

Response: We thanked the reviewer for pointing out this error. The first “HvOSCA2.1_2” should be revised to HVOSCA2.1_1. We have revised it and also carefully checked the contents throughout the manuscript.

4.In FIG2A, different colors to different clades gene name can help reader distinguish phylogenetic relationship.

Response: We appreciated the reviewer’s constructive and insightful suggestion. Following the suggestion, we have colored the members of HvOSCAs in each clade with different background colors. 

5.The number of OSCA genes described in line 267 is not consistent with that in Figure 5. And in Figure 5, what do the blue dots represent? The meaning and function of Figure 5 should be explained in more detail

Response: We appreciated the reviewer’s constructive and insightful comments. Actually, the number of OSCA genes described in line 267 were the HvOSCA involved in the co-expression network and 12 out 13 identified  HvOSCAs were found in 7 co-expression modules with 1, 1, 4, 1, 2, 1 and 1 HvOSCAs were found in the cyan, brown,blue, turquoise, green, pink, red modules, respectively.   The Figure 5 represented the integration of miRNA targeted HvOSCAs and co-expression network, which included 10 HvOSCAs targeted by 13 miRNAs and 16 miRNA-HvOSCA interactions. We are so sorry for this confusion. We have carefully revised it to make it clear and the meaning of Figure 5 was also further explained in more detail.

Reviewer 3 Report

First of all congrats for good work to the author and team. But there are a few suggestions from the reviewer in attached file. Hope for an improved version of manuscript. 

Author Response

First of all, congrats for good work to the author and team. But there are a few suggestions from the reviewer in attached file. Hope for an improved version of manuscript. 

Response: We appreciated the reviewer’s constructive suggestions and great help on language improvement. We have corrected spelling, grammar and other errors directly on the manuscript according to your comments. The additional comments reply as follows:

Commented [A61]: Please rewrite this line - here author wrote colors name - while those were new to this analysis then they were unable to understand so please revise with inference of these color what its meaning as a result

Response: We appreciated the reviewer’s constructive suggestion. We have rewritten the sentence.

Commented [A69]: Please rewrite the sentence as inference is misleading - what does author mean with litter

Response: We have revised it accordingly.

Commented [A90]: Please rewrite the sentence - exon-intron was investigated but what was outcome in this study

Response: We appreciated the reviewer’s constructive suggestion. We have rewritten the sentence to make it more informative.

Commented [A100]: Please rewrite this line - as inference is missing or misleading

Response: We appreciated the reviewer’s constructive suggestion. We are so sorry for these typos and we have carefully corrected the spelling and typos throughout the manuscript.

Commented [A97]: Please rewrite the sentence as inference is missing or misleading

Response: We appreciated the reviewer’s constructive suggestion. We have rewritten the sentence.

Commented [A101]: Please add the duration of experimentation in month and year duration. Also add the geographic coordinates (latitude, longitude and altitude) of experimental site and also the institute name where experiment was conducted

Response: We appreciated your constructive advice, we have added the details of the experiment, including the month and year duration and the experiment site.

Commented [A102]: Please add the source of variety - germplasm institute name (so that in future if any researcher works on similar research then can access the material.

Response: We are grateful to the reviewer for pointing out this problem. We have detailed the source of variety and make it clear.

Commented [A103]: Please add the reference of this protocol Commented

Response: We have fixed it.

Commented [A104]: Please add the reference of this protocol

Response: We have fixed it.

Commented [A105]: Please add the reference of this protocol

Response: We have fixed it.

Round 2

Reviewer 1 Report

I am happy for the authors' responses, and reccommend to acceptance for publication. 

Author Response

I am happy for the authors' responses, and reccommend to acceptance for publication. 

Response: We are grateful to reviewer for reviewing the manuscript again and give a positive approval towards our work

Reviewer 3 Report

Appreciate the improved manuscript. There are a few minor suggestions. Hope an improved version. 

Author Response

Appreciate the improved manuscript. There are a few minor suggestions. Hope an improved version.  

Response: We appreciated the reviewer’s constructive suggestions and great help in revising the manuscript. We have revised the manuscript according to your comments, and carefully checked the spelling and grammar in the article.